# Explanation Selection Using Unlabeled Data for Chain-of-Thought Prompting

**Xi Ye** and **Greg Durrett**
Department of Computer Science
The University of Texas at Austin
{xiye,gdurrett}@cs.utexas.edu

## Abstract

Recent work has shown how to prompt large language models with explanations to obtain strong performance on textual reasoning tasks, i.e., the chain-of-thought paradigm. However, subtly different explanations can yield widely varying downstream task accuracy. Explanations that have not been "tuned" for a task, such as off-the-shelf explanations written by non-experts, may lead to mediocre performance. This paper tackles the problem of how to optimize explanation-infused prompts in a black-box fashion. We first generate sets of candidate explanations for each example in the prompt using a leave-one-out scheme, then find an effective *combination* of these explanations with a two-stage framework. We first evaluate explanations for each in-context example *in isolation* according to two proxy metrics, log likelihood and accuracy on new examples. Then, we search over combinations of explanations to find one that yields high performance against a silver-labeled development set. Across four textual reasoning tasks spanning question answering, mathematical reasoning, and natural language inference, results show that our proxy metrics correlate with ground truth accuracy and our overall method can effectively improve prompts over crowdworker annotations and naive search strategies.[1]

## 1 Introduction

Large language models (LLMs) (Brown et al., 2020; Chowdhery et al., 2022) can be applied in various ways to do in-context learning (ICL). One line of work shows including *explanations* can boost the prompting performance on a diverse of reasoning tasks (Nye et al., 2021; Wei et al., 2022; Lampinen et al., 2022).[2] Despite the utility of such

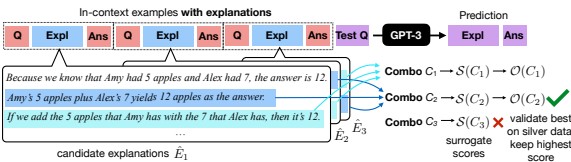

Figure 1: Optimizing explanations given a candidate set. We generate candidate explanations in a leave-one-out fashion (not shown), prioritize combinations of explanations using a surrogate score $\mathcal{S}$, then evaluate them on silver data to optimize accuracy.

explanations, they often require manual engineering (Wei et al., 2022; Zhou et al., 2022a) to reach their full potential; past work has demonstrated that different combinations of explanations can lead to widely varying model performance (Ye and Durrett, 2022; Wang et al., 2022b). Furthermore, these explanations are typically written in natural language (Madaan and Yazdanbakhsh, 2022; Ye et al., 2023; Wang et al., 2023) and there are naturally many variants to explain the answer to a single question. Explanations in standard datasets written by crowdworkers may not be optimal, and even expert "prompt engineers" may not be able to easily elicit the best behavior.

This paper studies the problem of optimizing explanations for better downstream performance on textual reasoning tasks. Inspired by recent work that bootstraps LLMs to improve reasoning (Zelikman et al., 2022; Huang et al., 2022), we propose an approach that can bootstrap a set of seed explanations (e.g., crowdworker annotated explanations) using an unlabeled development data set. As shown in Figure 1, we first prompt LLMs to construct alternative candidate explanations from the seed explanations. We then search over possible combinations of candidate explanations to find a combination that has high accuracy on the development set, which is silver-labeled using seed explanations.

Evaluating one candidate combination of explanations requires inference over the development set

---

[1]Code: https://github.com/xiye17/ExplSelection.

[2]Our paper uses the general term *explanation* to denote both chain-of-thought demonstrations for multi-step reasoning tasks as well as rationales for tasks like commonsense question answering, which do not involve chains of intermediate steps in the same way.

to compare against the silver labels. Given the cost of running LLMs, evaluating a large number of candidates is impractical. We propose a two-stage approach to efficiently search over potentially high-scoring combinations. We first evaluate each candidate explanation *in isolation* based on silver accuracy on the development set or the log likelihood on the few-shot training exemplar set. Scores of these individual explanations can be combined to compute scores of combinations, which gives a proxy of that combination's performance against silver set. We then can allocate our computation budget to evaluate better-performing candidate combinations based on the proxy metrics.

We apply our approach to optimize explanations on four datasets: GSM, ECQA, E-SNLI, and STRATEGYQA, covering a spectrum of textual reasoning tasks. Across the four datasets, our approach is able to find explanations that achieve 4% higher accuracy on average compared to initial seed explanations. We also show our proxy metrics can effectively approximate the downstream performance of combinations, and thus allow prioritizing search over better-performing explanations.

To summarize, our contributions are: (1) We propose a framework for optimizing explanations for in-context learning by optimizing over combinations of explanations. (2) We show that pseudo-labeling an unlabeled dataset can be used to evaluate such combinations. (3) We propose two proxy metrics to prioritize exploring better combinations given a limited computation budget.

## 2 Problem Formulation

### 2.1 Problem Statement

Following the standard chain-of-thought setting (Wei et al., 2022), we assume access to a set of *exemplars* (input-output pairs) $T = \{(q_i, a_i)\}_{i=1:K}$ and *seed explanations* $\tilde{E} = \{\tilde{e}_i\}_{i=1:K}$ annotated for each exemplar in $T$ (one per exemplar). In addition to $T$, some of our approaches assume access to an *unlabeled development set* $V$ that only includes the inputs, i.e., $V = \{q_i\}_{i=1:M}$. Let $\theta$ be the parameters of an LLM.

Our goal is to find an explanation set $E = \{e_i\}_{i=1:K}$ that leads to the best accuracy. Each $e_i \in \Sigma^*$ is a natural language explanation expressed in the subword vocabulary $\Sigma$ of the pre-trained language model. Past work has optimized many aspects of the in-context learning process, for example, the verbalization of prompts (Deng et al.,

2022; Zhang et al., 2022), exemplar selection (Ye et al., 2023), and exemplar order (Lu et al., 2022), whereas our work focuses on optimizing the format of explanations in this particular way.

Because we assume a very small number of training examples, all of which are going to be included in the prompt, our notion of optimization (our "training objective") cannot rely on maximizing the likelihood of labeled training data. As we discuss in future sections, we will explore both likelihood-based measures as well as accuracy against pseudo-labeled versions of $V$. These objectives are also expensive to evaluate using LLMs, so we will operate under an additional constraint of cost in our methods.

**Candidate explanations** Directly searching over the combinatorial explanation space of $E$ is intractable. Practically, we constrain the space of each $e_i$ by selecting each from a *candidate explanation set* $\hat{E}_i = \{\hat{e}_i^{(1)} \ldots \hat{e}_i^{(|\hat{E}_i|)}\}$, where each $\hat{e}_i^{(j)}$ denotes a candidate explanation associated with each exemplar $q_i$. The candidate explanation sets $\hat{E}_1 \ldots \hat{E}_K$ can be generated by the LLM using a set of manually annotated seed explanations annotated by human $\tilde{E} = \{\tilde{e}_i\}_{i=1:K}$. That is, we use the exemplar set $T$ and the seed sets $\tilde{E}$ excluding $(q_i, \tilde{e}_i, a_i)$ to prompt the LLM and draw $N$ (40 in our implementation) samples for $\hat{E}_i$:

$$(\hat{e}, \hat{a}) \sim p(e, a_i \mid \{(q_j, \tilde{e}_j, a_j)\}_{j=1:K \land j \neq i}, q_i; \theta) \quad (1)$$

Put another way, we use a leave-one-out approach to sample explanations and answers for each example using chain-of-thought prompting with $K - 1$ examples. We reject any samples that do not have the correct answer for the example.

A combination $C$ is a set of $\{e_i\}$ that contains one explanation $e_i$ from the candidate explanation set $\hat{E}_i$, i.e., $C = \{e_i\}_{i=1:K} \land \forall i, e_i \in \hat{E}_i$. Now we can restate our problem: our goal is to find an explanation combination $C$ that maximizes the accuracy when evaluating on test data.

### 2.2 Performance Varies Across Explanations

To illustrate the potential of our approach, we briefly analyze how using different explanations, for the same set of exemplars, can impact the downstream performance. As mentioned earlier, we generate candidate explanation sets according to Eq (1). Concretely, we use temperature scaling of 0.7 and sample 40 completions for each $q_i$, only retaining

|  | Min | Avg | Max | Seed |
|---|---|---|---|---|
| GSM | 57.7 | 61.8 | 66.0 | 61.9 |
| ECQA | 72.7 | 76.1 | 78.6 | 74.9 |
| E-SNLI | 60.3 | 72.3 | 80.1 | 71.8 |
| STRATEGYQA | 69.8 | 73.8 | 76.5 | 74.0 |

Table 1: Statistics of the performance of 16 different random combinations of explanations on 4 datasets and the performance of the seed explanations from crowdworkers. All tasks show substantial variation in performance.

an $\bar{e}$ if it is paired with a correct answer $\bar{a} = a_i$. Note that for different $q_i$, we may find varying number of valid $\bar{e}$ (ranging from 0 to 40). We keep at most 8 for each $q_i$ to save the search cost. We also include the seed explanations in the candidate explanation sets.

For each dataset, we randomly sample 16 combinations using the augmented candidate explanation sets, and report the statistics of the performance in Table 1. We see substantial variance in performance with different $C$: the average gap between the maximum performance and minimum performance exceeds 5% and is as large as 20% (on E-SNLI). In addition, the performance of seed explanations annotated by crowdworkers (SEED in Table 1) largely lags the best possible explanations, indicating substantial headroom for improvement.

## 3 Method Overview

Having candidate explanations for each question, we have reduced the search space from exponential in the vocabulary size to merely $N^K$. We then search over possible combinations of explanations. We describe our method for scoring combinations and the constraints under which our search takes place.

**Pseudo-labeling development set** We do not assume access to labeled examples beyond the $K$ few-shot examples provided. However, we can take advantage of unlabeled data in $V$. We use a *pseudo-labeling* approach to derive labels for $V$ following past work (Wang et al., 2022c). This approach is depicted in Figure 2; given $q \in V$, we sample random combinations of explanations to get predictions and use the majority-voted answer as the pseudo label $\hat{a}$:

$$\hat{a} = \arg\max_a \sum_{C=\{e_i\}} \mathbb{1}[a = \arg\max_{\bar{a}} p(\bar{a} \mid \{(q_i, e_i, a_i)\}_{i=1:K}, q; \theta)] \quad (2)$$

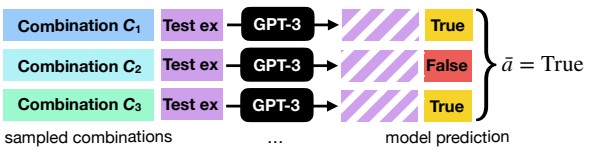

Figure 2: Silver labeling of unlabeled test example given several sampled combinations. This example is for a binary task with True or False labels (e.g., StrategyQA).

We now use the accuracy against the silver label as a surrogate objective $\mathcal{O}$, searching for $C$ that maximizes accuracy with respect to the $\hat{a}$:

$$\mathcal{O}(C) = \arg\max_{C=\{e_i\}_{i=1:K}} \sum_{q_j \in V} \mathbb{1}[\hat{a}_j =$$
$$\arg\max_{\bar{a}} p(\bar{a} \mid \{(q_i, e_i, a_i)\}_{i=1:K}, q_j; \theta)]. \quad (3)$$

**Searching over combinations** One further complicating factor is that evaluating a combination $C$ using $\mathcal{O}$ is expensive, as it requires running inference over the development set. We measure the computation budget $B$ by the number of combinations needed to be scored using $\mathcal{O}$.

A naive approach is to randomly select $B$ combinations to search, but this is inefficient. We propose additional surrogate metrics $\mathcal{S}$ to serve as a proxy for $\mathcal{O}$ for scoring combinations. We design $\mathcal{S}$ so that it can cost-efficiently score all combinations, with high $\mathcal{S}(C)$ indicating a combination $C$ likely to obtain high $\mathcal{O}(C)$ score. In this way, $\mathcal{S}$ can be used to propose promising candidate combinations, only a few of which are scored using the actual objective $\mathcal{O}$ to save search budget.

## 4 Proxy Metrics for Finding Promising Combinations

Owning to the high cost, we only evaluate a small number (tens of combinations) of combinations against development set using $\mathcal{O}$ (Eq (3)). We first extract a set of promising combinations according to two proxy metrics, then evaluate those using our silver data.

### 4.1 One-shot Silver Accuracy

To optimize the silver accuracy of a combination of explanations (our objective $\mathcal{O}$), we hypothesize that *the prediction of a combination can be approximated with the prediction of each explanation used one-shot.* That is, we expect $p(a \mid \{(q_i, e_i, a_i)\}_{i=1:K}, q; \theta)$ to be higher when

$\sum_{i=1:K} p(a \mid (q_i, e_i, a_i), q; \theta)$ is higher. We draw this hypothesis based on recent work on example selection for ICL, which shows that combining examples that individually perform well will yield better performance from the combination (Ye et al., 2023; Rubin et al., 2022).

We define the average one-shot silver accuracy as a proxy metric $\mathcal{S}_{\text{OSAcc}}$:

$$\mathcal{S}_{\text{OSAcc}}(C = \{e_i\}_{i=1:K}) = \sum_{i=1:K} \sum_{q_j \in V} \mathbb{1}[\hat{a}_j = $$
$$\arg\max_{\bar{a}} p(\bar{a} \mid (q_i, e_i, a_i), q_j; \theta)] \quad (4)$$

By computing the one-shot silver performance for $\forall \hat{e}_j^{(i)} \in \hat{E}^{(i)}$ for $\forall i = 1 : K$, we can efficiently compute the proxy metric $\mathcal{S}_{\text{OSAcc}}$ for any combination $C$.[3]

## 4.2 One-shot Log Likelihood

Besides using silver accuracy, another principle is to optimize the held-out log likelihood of the exemplar set:

$$\sum_{j=1:K} \log p(a_j \mid \{(q_i, e_i, a_i)\}_{i=1:K \wedge i \neq j}, q_j; \theta).$$

We apply a similar hypothesis and use the one-shot performance $\sum_{i=1:K \wedge i \neq j} p(a_j, \mid (q_i, e_i, a_i), q_j; \theta)$ as the surrogate of $p(a_j \mid \{(q_i, e_i, a_i)\}_{i=1:K \wedge i \neq j}, q_j; \theta)$. We can then score a candidate combination by:

$$\sum_{j=1:K} \sum_{i=1:K \wedge i \neq j} \log \sum_e p(a_j, e \mid (q_i, e_i, a_i), q_j; \theta).$$

Since summing over explanations is intractable, we approximate this sum using the single sample of $e$ to estimate the one-shot performance, leading to:

$$\mathcal{S}_{\text{OSLL}} = \sum_{j=1:K} \sum_{i=1:K \wedge i \neq j} \log p(e_j, a_j \mid (q_i, e_i, a_i), q_j; \theta).$$
$$(5)$$

We can compute $\mathcal{S}_{\text{OSLL}}$ for any $C$ by only computing all the pairwise probabilities, $p(e_j, a_j \mid (q_i, e_i, a_i), q_j; \theta)$, for $\forall e_i \in \hat{E}_i, e_j \in \hat{E}_j \forall i = 1 : K, j = 1 : K \wedge i \neq j$, which is computationally feasible. Note that this metric does not require a development set.

---

[3]While this involves $NK$ evaluations on the silver set, note that these evaluations are one-shot and significantly less computationally expensive than using higher numbers of shots.

## 4.3 Ensemble of $\mathcal{S}_{\text{OSAcc}}$ and $\mathcal{S}_{\text{OSLL}}$

We have described the two proxy metrics using either the unlabeled set $V$ or the labeled few-show exemplars $T$. Our further analysis (which we will describe later in Section 4) shows the choice of the most effective metric is task-specific. We additionally propose a strategy, ENSEMBLE of the $\mathcal{S}_{\text{OSLL}}$ and $\mathcal{S}_{\text{OSAcc}}$. Specifically, we first construct two sets of combinations that are preferred by these two proxy metrics individually, and then select the best one, from the union of these two sets, according to $\mathcal{O}$.

## 5 Experimental Setup

### 5.1 Language Models

We primarily use code-davinci-002 (Chen et al., 2021), a state-of-the-art LLM API, throughout our experiments, given its strong performance on various reasoning tasks (Li et al., 2022b). In addition, we use text-davinci-003 to verify the effectiveness of the proxy metrics. code-davinci-002 is a base model, and text-davinci-003 is an Instruct-series model fine-tuned to align with human preferences (Ouyang et al., 2022).

**Inference** We follow past work to employ *greedy decoding* (greedily selecting the most probable token autoregressively) (Wei et al., 2022; Ye and Durrett, 2022) or self-consistency decoding (sampling tens of outputs from LLMs via temperature scaling and using popularity voting to assign a label) (Wang et al., 2022c).

**Cost** Querying LLMs is computationally intensive. We aim to search for better explanations within a reasonable budget. Our evaluation of cost is based on the *number of tokens* processed by LLMs, including both tokens in the prompts and the tokens generated by LLMs. We further bucket the measurement of cost by the number of combinations $C$ that are scored by $\mathcal{O}$, which involves processing $M(K + 1)$ examples.

### 5.2 Datasets

We experiment with four datasets covering four distinct tasks, including:

- GSM (Cobbe et al., 2021) consists of grade school math questions. Each is paired with a human-written explanation for the answer.

| METRICS | GSM | | ECQA | | ESNLI | | STRATEGYQA | |
|---|---|---|---|---|---|---|---|---|
| | Max@8 | Max@16 | Max@8 | Max@16 | Max@8 | Max@16 | Max@8 | Max@16 |
| NAIVE | 65.1 | 66.0 | 78.6 | 78.6 | 79.5 | 80.1 | 76.2 | 76.5 |
| $\mathcal{S}_{OSAcc}$ | **66.4** | **67.0** | 79.7 | 80.5 | **80.4** | **81.2** | 74.3 | 74.9 |
| $\mathcal{S}_{OSLL}$ | 65.7 | 65.9 | **80.2** | **80.6** | 75.8 | 76.5 | **77.1** | **77.4** |

Table 2: Oracle maximum accuracies achievable with 8 or 16 candidate combinations using different selection strategies. Using log likelihood-based or silver accuracy-based proxy metrics can find more promising candidate combinations than random candidates.

- ECQA (Aggarwal et al., 2021; Talmor et al., 2019) contains multiple-choice questions which test models' commonsense knowledge.

- E-SNLI (Camburu et al., 2018) studies the task of natural language inference which is to classify the relation between a premise and a hypothesis.

- STRATEGYQA (Geva et al., 2021) asks Yes-No questions requiring steps. The dataset does not have explanation annotations, but it provides facts (Geva et al., 2021) which are supporting evidence (albeit noisy ones) for the answers, so we use them as explanations.

For each of the datasets, we choose prompt formats commonly used in past work (Wei et al., 2022; Wang et al., 2022b). We show one example in the corresponding prompt format in Appendix A. We use 8 exemplars in prompts for GSM, ECQA, and STRATEGYQA, and 9 exemplars (3 for each class) for E-SNLI, as sing more exemplars would not lead to further performance gains.

## 6 Effectiveness of Proxy Metrics

Before showing the results of the complete system, we first present experiments for verifying the effectiveness of the two proxy metrics. We evaluate them on the basis of the best oracle accuracy on a small (gold) labeled test set that we can reach using the top-$X$ candidates, referred to as MAX@$X$, ranked by $\mathcal{S}_{OSAcc}$ or $\mathcal{S}_{OSLL}$. This gives an oracle upper bound for the performance that silver reranking via $\mathcal{O}$ can yield.

**Setup** We compare our metrics against a baseline which randomly scores combinations (NAIVE). We mainly use `code-davinci-002` for this experiment; please refer to Appendix B for additional results on `text-davinci-003`. For $\mathcal{S}_{OSAcc}$, we silver-labeled 256 randomly drawn development with 48 samples of combinations. For each dataset, we experiment with four different exemplar sets $T$ to control for randomness and report the average number.

**Results** Table 2 shows the maximum reachable performance within 8 (Max@8) and 16 (Max@16) candidate combinations. For each dataset, using one of our metrics can find more promising candidate combinations than randomly proposed candidates. Among the top 16 combinations, combinations preferred by $\mathcal{S}_{OSAcc}$ can achieve better performance than randomly selected combinations by 1.0%, 0.9%, and 1.4% on GSM, ECQA, and E-SNLI, respectively. $\mathcal{S}_{OSLL}$ is the most effective strategy on ECQA, and STRATEGYQA, surpassing NAIVE by 2.0% and 0.9% on the basis of 16 candidate combinations. We do not find one metric that consistently gives the best performance.

**Proxy metrics vs downstream accuracy** In Figure 3, we show a series of graphs for intuitive understanding of how the proxy metrics relate to the downstream accuracy. Each group of graphs shows the downstream accuracy vs. the surrogate proxy scores of combinations preferred by different metrics. For each dataset, we show two groups of graphs for two different exemplar sets out of four. Each group contains three graphs with different values on the x-axis. The first graph of a triple shows $\mathcal{S}_{OSAcc}$ on the x-axis and the second one shows one-shot likelihood on the exemplar set (positively correlates with $\mathcal{S}_{OSLL}$). In addition to the two proxy metrics, we show the completion likelihood on the third graph (probability of the predictions on the development set).

We show that the two surrogate scores we define mostly positively correlate with the downstream accuracy. $\mathcal{S}_{OSAcc}$ (left) works uniformly well except on STRATEGYQA. $\mathcal{S}_{OSLL}$ works well except for Figure 3a from GSM and Figure 3f from E-SNLI. In particular, on ECQA, both of them highly positively correlate with the downstream accuracy. Furthermore, we show the candidate combinations preferred by our proxy metrics lead to, in most

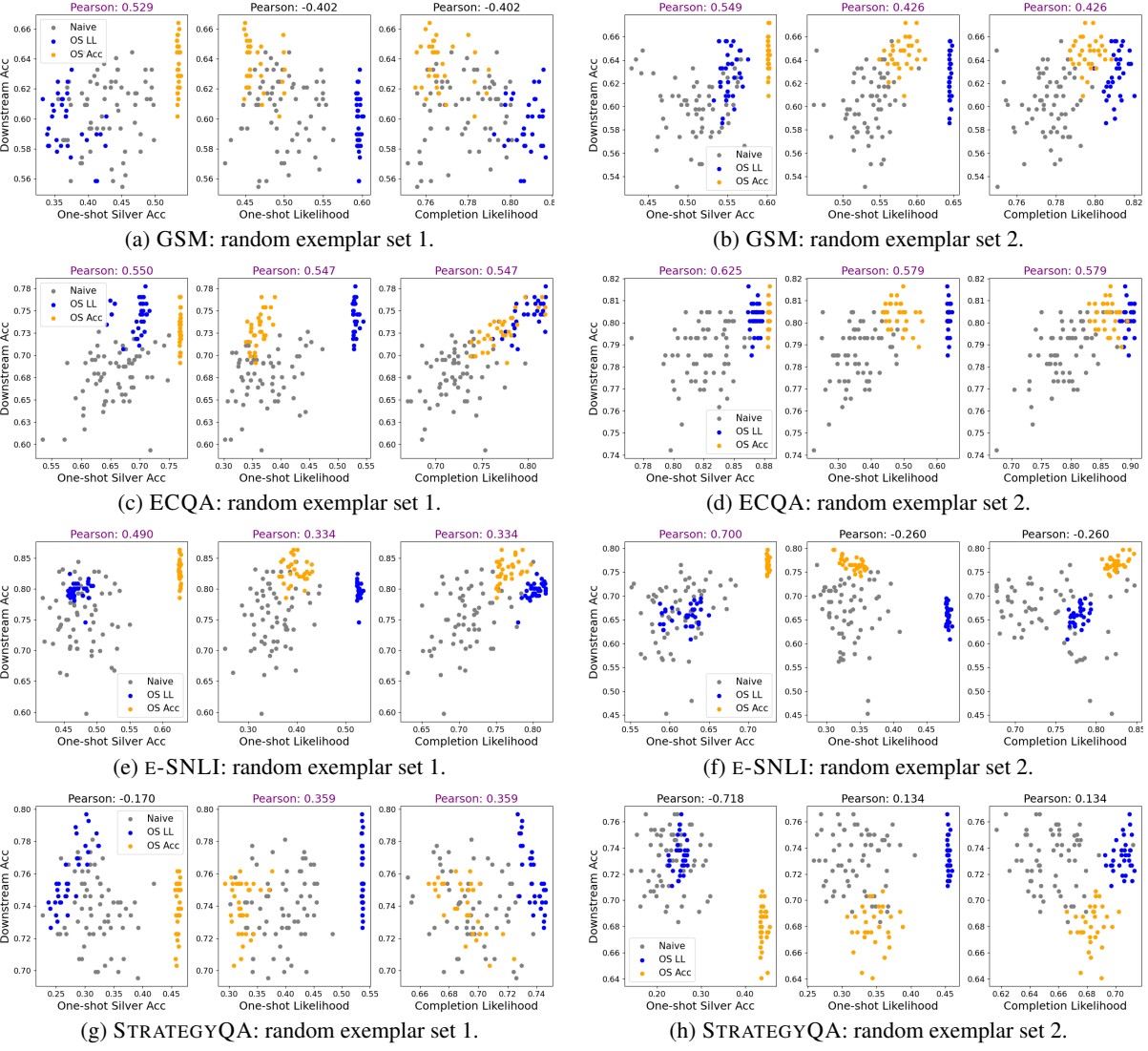

Figure 3: Gold test set accuracy (y-axis) vs. various surrogate proxy scores for explanation sets. Points of three different colors denote combinations selected using three metrics. There is a positive correlation between $\mathcal{S}_{\text{OSAcc}}$ and performance on these datasets except for STRATEGYQA (Pearson above 0.3 is highlighted in purple). $\mathcal{S}_{\text{OSLL}}$ also shows positives correlation on ECQA and STRATEGYQA and occasionally fails on the others.

cases, better likelihood on the development set (third graph in each triple), which indicates these combinations are more "optimized" for a specific task; past work suggests that better likelihood generally correlates with better downstream performance (Gonen et al., 2022).

## 7 Effectiveness of Framework

### 7.1 Main Results

We now test the effectiveness of the full framework. We mainly compare the performance of the explanations optimized via our approach against (1) the ZERO-COT approach (Kojima et al., 2022) (not using any human provided explanations) and (2) using seed explanations. In addition, we derive two

baselines from past work on constructing effective explanations for ICL, which also select potentially better explanations from candidate explanations. Recall that $\hat{E}_i = \{\hat{e}_i^{(1)} \dots \hat{e}_i^{(|\hat{E}_i|)}\}$ is the candidate explanation set for $q_i$, our baselines include (1) BESTLEN that chooses the longest explanations (i.e., $\max_{\tilde{e} \in \tilde{E}} |\tilde{e}|$), as Fu et al. (2022) suggest using more complex CoTs leads to better performance for arithmetic reasoning, and (2) BESTPPL that chooses the explanation with the best perplexity (i.e., $\max_{\tilde{e} \in \tilde{E}} \text{Perplexity}(a_i, \tilde{e}, q_i)$), as Gonen et al. (2022) suggest lower perplexity of prompts correlate with better performance. We note that these two baselines are not invented for optimizing explanations of given exemplars and are adapted to fit

| | GSM | ECQA | E-SNLI | STRQA |
|---|---|---|---|---|
| ZERO-COT | 30.9 | 61.2 | 49.7 | 55.1 |
| SEED | 62.6 | 77.0 | 75.2 | 71.3 |
| BESTLEN | 61.8 | 74.6 | 74.9 | 68.3 |
| BESTPPL | 63.4 | 79.4 | 76.5 | 69.0 |
| **OPTIMIZED** | **66.0** | **83.0** | **82.8** | **71.6** |

Table 3: The performance of optimized explanations against seed explanations and baselines derived from past work. Optimized explanations substantially outperform other approaches on GSM, ECQA, and E-SNLI.

| NUM | EXPL | GSM | ECQA | E-SNLI | STQA |
|---|---|---|---|---|---|
| 5 | SEED | 70.4 | 79.8 | 80.0 | 72.9 |
| 5 | OPTIM | 73.5 | 81.5 | 85.1 | 71.9 |
| 10 | SEED | 74.9 | 81.1 | 82.5 | 73.5 |
| 10 | OPTIM | 78.9 | 82.1 | 85.5 | 73.1 |
| 20 | SEED | 79.1 | 81.2 | 83.7 | 74.4 |
| 20 | OPTIM | 80.5 | 82.5 | 86.3 | 74.0 |
| 40 | SEED | 80.1 | 81.5 | 84.6 | 75.0 |
| 40 | OPTIM | 81.2 | 82.5 | 87.2 | 75.4 |

Table 4: Performance of seed explanations and optimized (Optim) explanations using self-consistency decoding with varying number of samples.

| | GSM | ECQA | E-SNLI | STRQA |
|---|---|---|---|---|
| SEED | 58.2 | 74.3 | 81.0 | 67.6 |
| OPTIMIZED | 61.3$^{\Uparrow}$ | 76.9$^{\Uparrow}$ | 82.8$^{\uparrow}$ | 69.4$^{\Uparrow}$ |

Table 5: The performance of optimized explanations against seed explanations on text-davinci-003 ($\Uparrow$ and $\uparrow$ denote significant improvements with p < 0.05 and p < 0.1, respectively). Our optimization approach is effective across LLMs.

our setting. We refer to our optimization approach (based on the ENSEMBLE strategy) as OPTIMIZED.

**Setup** For all dataset sets, we experiment with 4 different exemplar sets as well as different unlabeled sets $V$ of 256 randomly selected examples. We sample 48 combinations to silver label $V$. We constrain the computation budget $B$ to be 50; this was the highest point feasible given limitations and was also where we found the silver accuracy ($\mathcal{O}$) to be nearly saturated. We note this budget has included the overhead for computing the proxy metrics as well as the computation for scoring combinations using $\mathcal{O}$ (see Appendix C for details).

**Results** We show the performance of different approaches in Table 3. Overall, using our framework can find substantially better explanations measured by prompting performance compared to seed explanations. Without using any manually annotated explanations, the performance of ZERO-COT is far behind few-shot prompting using the seed explanations (SEED). Meanwhile, the explanations optimized using our framework outperforms the original seed explanations by 3.3%, 4.3%, and 7.1%, on GSM, ECQA, and E-SNLI, respectively. Choosing explanations with the lowest perplexity (BESTPPL) is able to marginally improve the performance on GSM, ECQA, and E-SNLI, compared to the seed set, but is consistently worse than our approach, and even leads to performance degradation on STRATEGYQA. As we are using 4 different random exemplar sets, we perform 4 groups of significance tests for different random trials. We note the gain of our approach over the seed set is typically significant, please refer to Appendix F for details.

## 7.2 Analysis

**Self-consistency performance** In addition to greedy decoding used in Table 3, we evaluate the performance of our optimized explanations under self-consistency decoding and compare against seed explanations. We vary the number of samples from 5 to 40, and show the results in Table 4. We note that the results are on a basis of one random exemplar set for each of the datasets, owing to the high computational cost of drawing tens of samples. As shown in Table 4, the optimized explanations consistently outperform the seed explanations under different numbers of samples. The gap is especially significant with smaller number of samples.

**Results on other LLMs** We mainly uses code-davinci-002 in our experiments given its state-of-the-art ICL abilities. We also verify the effectiveness of our approach on text-davinci-003, an LLM finetuned to align with human feedback (Ouyang et al., 2022). We note that experiment with a smaller scale given the high cost (see Appendix B for details) and evaluate on one random set of exemplars instead of four. As shown in Table 5, applying our approach can also find better-performing explanations for all the datasets on text-003. Analysis on the effectiveness of our proxy metrics on text-003 is also included in Appendix B.

|            | SVAMP | SINEQ | SINOP | ADDSUB | MULARI |
|------------|-------|-------|-------|--------|--------|
| SEED       | 73.0  | 92.8  | 91.5  | 86.7   | 95.0   |
| OPTIM-GSM  | 76.9  | 93.4  | 92.2  | 89.6   | 95.6   |

Table 6: Explanations optimized on the GSM dataset (OPTIM-GSM) achieve better performance on SVAMP and different settings of MAWPS compared to the seed explanations. The performance improvements of optimized explanations on one dataset can generalize to other out-of-domain datasets.

|           | GSM  | ECQA | E-SNLI | STRQA |
|-----------|------|------|--------|-------|
| SEED      | 62.6 | 77.0 | 75.2   | 71.3  |
| OPTIMIZED | 64.5 | 81.2 | 81.5   | 71.0  |

Table 7: Results of searching with a reduced budget. Optimized explanations can still improve the performance upon the seed explanations.

**Generalizability of optimized explanations** We investigate whether the performance improvements of our optimized explanations in a particular domain can generalize to other datasets with different distributions. Table 6 shows the performance of seed explanations and the optimized explanations from the GSM dataset (OPTIM-GSM) on the other arithmetic reasoning datasets, including SVAMP (Patel et al., 2021) and MAWPS (Koncel-Kedziorski et al., 2016). As suggested by the results, the optimized explanations achieve better performance compared to seed explanations on the out-of-domain datasets, which indicates that the performance improvements can generalize.

**Results with reduced computation budget** We expect search with our proxy metrics can still work well without high computation budget since they already extract potentially high-scoring combinations. We test a setting that uses a reduced computation budget. We set the budget to be 20 (as opposed to 50 in the main experiments; see Appendix C for more details). As seen in Table 7, with reduced budget, our framework can still improve the downstream performance compared to seed explanations by around 2.0%, 4.0%, and 6.0%, on GSM, ECQA, and E-SNLI, while maintaining performance on STRATEGYQA.

**Failure analysis of proxy metrics** In Section 6, we see that the $\mathcal{S}_{OSLL}$ and $\mathcal{S}_{OSAcc}$ do not always positively correlate with the performance on certain datasets. While we show such uncertainty can be handled by using an ensemble and scoring based on

$\mathcal{O}$ we briefly analyze the failure of the two metrics for a better understanding of them.

In Table 2, $\mathcal{S}_{OSAcc}$ performs poorly on STRATEGYQA, yielding lower performance than the NAIVE strategy. The silver accuracy on this dataset is very poor: almost all one-shot accuracy is below 50% (see Figure 3g), worse than random guessing. One reason is that the binary nature of the task causes a single demonstration to be less suitable and representative than a single demonstration on more complex tasks like GSM. Under such circumstances, the averaged one-shot accuracy is no longer indicative of the full-prompt silver accuracy. On the other datasets, one-shot accuracy is meaningful (better than random guess), and the $\mathcal{S}_{OSAcc}$ correlates well with the full-prompt accuracy.

Furthermore, combinations scored highly by $\mathcal{S}_{OSLL}$ in Figure 3f are not better than random combinations in terms of downstream accuracy. Such combinations also lead to a mediocre completion likelihood, which is unusual as optimizing $\mathcal{S}_{OSLL}$ typically leads to the highest completion likelihood in other cases in Figure 3. We hypothesize this can be attributed to the distribution gap between the exemplar set and the test set. Since $\mathcal{S}_{OSLL}$ optimizes the log likelihood only based on the exemplar set, it might not generalize well to the test set under severe distribution shift, which is indicated by the suboptimal completion likelihood.

**Analysis on proxy metrics** In Section 6, we investigate the effectiveness of our proxy metrics with the oracle accuracy on a small test set. We provide additional analysis on proxy metrics in Appendix D, which shows applying our approach in a naive way (without using proxy metrics) can already lead to accuracy improvements compared to the seed set, using proxy metrics to prioritize search strategy can further improve the performance of the searched explanations.

**Output examples** We include examples of the original explanations and the search outputs in Appendix G. We note that not all optimized explanations necessarily look much better or more plausible as perceived by humans. The optimization objective here is designed to induce better test predictions in the final model. Part of the effects of this optimization may also be in the combination of the different explanations, so explanations may also be selected because they are more "compatible" with others in the final $\mathcal{O}$ ranking function.

## 8 Related Work

We study prompting LLMs with chain-of-thought (Nye et al., 2021; Wei et al., 2022; Shi et al., 2022) or textual explanations more generally (Marasović et al., 2022; Ye and Durrett, 2022). Much of the past work focuses on exemplar selection in the presence of explanations (Fu et al., 2022; Ye et al., 2023) or developing prompting methods for various reasoning tasks (Jung et al., 2022; Gao et al., 2022), which typically require manually engineered explanations. We focus instead on searching for better-performing explanations.

Our approach leverages data without explanation annotations. Similarly, prior work also explores the means of using few-show explanations together with data points without explanations annotations for improving downstream performance (Zelikman et al., 2022; Li et al., 2022b; Ye et al., 2023; Li et al., 2022a; Wang et al., 2022a; Huang et al., 2022). Many of these techniques need a large amount of fully labeled data to train models used for generating explanations (Zelikman et al., 2022) or smaller models used as verifiers (Li et al., 2022b,a; Wang et al., 2022a), whereas our work only uses a small unlabeled set. There is also work on automatically constructing CoTs (Zhang et al., 2023) starting ZoTs (Kojima et al., 2022), which also requires a fully labeled dataset. In particular, Huang et al. (2022) also use LLMs to silver labeled data points for finetuning the LLMs; our work instead treats LLMs as black-boxes and searches for better explanations instead of tuning the parameters.

Our work also closely relates to prompt optimization. While experts can potentially engineer better prompts (Reynolds and McDonell, 2021; Mishra et al., 2022), such a process requires heavy manual effort. This has attracted growing interest on automated prompt engineering. One line of work requires interacting with gradients (Shin et al., 2020; Hu et al., 2021) or continuous embeddings (Sun et al., 2022a,b; Diao et al., 2022; Sun et al., 2023). Another line uses LMs as black-boxes (Prasad et al., 2022; Deng et al., 2022; Zhang et al., 2022; Zhou et al., 2022b). However, this past work either optimizes over discrete templates (not applicable for the explanation optimization setting) or optimizes over string verbalizations (a search space too large for our setting).

## 9 Conclusion

We have presented an approach that can search for better-performing explanations for ICL starting from a set of seed explanations. Our approach first proposes promising candidate combinations of alternative explanations generated using LLMs, then finds explanation combinations using proxy metrics before using a silver-labeled validation set to select the best candidate. Our results highlight the substantial variance in the performance of different sets of explanations, paving the way for future work to further optimize explanations in this paradigm.

## Limitations

Our approach highly relies on the capabilities of the LLMs. We use LLMs to generate candidate explanations, to silver-label development set, as well as to score combinations. To that end, we hypothesize less capable LMs might see limited benefits from our approach, and it is more suitable in a setting that involves finetuning using a large number of labeled set (Zelikman et al., 2022).

Our approach requires overhead cost to optimize the explanations, including pseudo-labeling the development and scoring combinations using silver accuracy. However, at inference time, the cost is the same as standard few-shot prompting with explanations. We believe it is reasonable to pay a moderate "training" cost; if optimizing an LLM prompt that is to be deployed as a service, the cost at the training stage (equivalent to running self-consistency inference on 500 test examples) is acceptable compared to the long-term costs of running the model on examples.

Our approach optimizes the silver accuracy via searching over combinations preferred by proposed proxy metrics. This does not guarantee finding the combination with optimal silver accuracy, especially as we are limiting our computation budget and operating in the black-box setting. While there exist approaches that use gradient-based optimization for more exhaustively searching over a smaller set of options, (e.g., RLPrompt (Deng et al., 2022) searches over prompts that are just a few tokens long), we are not aware of any method that can search over the space of prompts for black-box LLMs and find a provably optimal prompt. Our trade-off reflects the practical constraints of this complex setting.

Our approach optimizes the downstream performance by optimizing explanations, leaving out

other factors such as verbalization and exemplar order. In particular, we find varying explanations grants more substantial headroom than varying order (see Appendix E for detailed discussion).

Lastly, this work only considers a certain range of reasoning datasets written in English. It is unknown how well our approach can handle other languages, or other reasoning tasks such as pure symbolic reasoning.

## Acknowledgments

Thanks to anonymous reviewers for their helpful feedback, as well as to Eunsol Choi, Chenglei Si, Qiaochu Chen, Huancheng Chen, Yasumasa Onoe, Jiacheng Xu, Jifan Chen, Zhen Chen, Yunmo Chen, and Lemeng Wu for their help with various aspects of this work. This work was supported by NSF CAREER Award IIS-2145280 and the NSF Institute for Foundations of Machine Learning.

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

## A  Datasets & Prompt Examples

We show an example and corresponding prompt format for each of the datasets we use in Figure 4.

| GSM |
| --- |
| Q: In a basketball game, Tobee scored 4 points. Jay scored 6 more than Tobee and Sean scored 2 less than the points of Tobee and Jay together. If Tobee, Jay, and Sean are on the same team, how many points did they score for their team?
A: Jay scored 4 + 6 = 10 points. Together, Tobee and Jay scores 4 + 10 = 14 points. So, Sean scored 14 - 2 = 12 points. Thus, Tobee, Jay, and Sean scored a total of 4 + 10 + 12 = 26 points for their team. The answer is 26. |

| ECQA |
| --- |
| Q: The child was spiteful of his parents, what did he do?
Answer Choices:
(a) become adult
(b) succeeded
(c) grow up
(d) ask questions
(e) acting out
A: Children act out. Acting out is a type of behaviour. Spiteful people act out. So the answer is (e). |

| E-SNLI |
| --- |
| Premise:
"A man at a flea market browsing."
Based on this premise, can we conclude the hypothesis "A man is sleeping at a flea market." is true?
OPTIONS:
- yes
- no
- not possible to tell
A: One cannot be sleeping and browsing at the same time. The answer is no. |

| STRATEGYQA |
| --- |
| Q: Did Archduke Franz Ferdinand of Austria participate in the Pacific War?
A: Archduke Franz Ferdinand of Austria was assassinated in 1914. The Pacific War took place between 1941 and 1945. So the answer is no. |

Figure 4: Examples of prompts for GSM, ECQA, E-SNLI, and STRATEGYQA.

## B  Experiments of the Effectiveness of Proxy Metrics on `text-davinci-003`

In addition to `code-davinci-002`, which we mainly use throughout the paper, we also verify the effectiveness of our proxy metrics on `text-davinci-003`. Unlike `code-002`, which is a based model, `text-003` is an instructional finetuned model (that learns to maximize a reward model trained from comparisons by humans).

| METRICS | GSM
MAX@8 | ECQA
MAX@8 | E-SNLI
MAX@8 | STRATEGYQA
MAX@8 |
| --- | --- | --- | --- | --- |
| NAIVE | 57.0 | 74.1 | 81.2 | 71.9 |
| $\mathcal{S}_{\text{OSAcc}}$ | 61.7 | 75.4 | 81.3 | 71.1 |
| $\mathcal{S}_{\text{OSLL}}$ | 56.3 | 75.8 | 80.9 | 72.5 |

Table 8: Oracle maximum accuracies achievable with 8 candidate combinations on `text-davinci-003`. The trend is similar to the results of `code-davinci-002`.

**Setup**    As in Section 6, we evaluate the maximum reachable performance within 8 (Max@8) candidate combinations. Given the cost for querying the API, we conduct experiments with a smaller scale: we only use 12 samples to silver-label development set, and evaluate on only one set of exemplars for each dataset.

**Results**    As shown in Table 8, we observe a similar trend to `code-davinci-002` which is used in Section 6: $\mathcal{S}_{\text{OSAcc}}$ is particularly effective on GSM and ECQA, whereas $\mathcal{S}_{\text{OSLL}}$ is effective on ECQA and

STRATEGYQA. We see somewhat larger gains on GSM (over weaker baseline performance) and less change in E-SNLI (over a stronger baseline model).

## C  Details of Computation Overhead and Computation Budget

**Details of computation overhead for proxy metrics**  We detail the computation overhead needed for $\mathcal{S}_{\text{OSAcc}}$ and $\mathcal{S}_{\text{OSLL}}$. Recall that we bucket the measurement of cost by the number of combinations $C$ that are scored by $\mathcal{O}$. Scoring one combination involves processing $M(K+1)$ examples (ruining inference $M$ data points with $K$ examples in prompts and 1 example in output), which we use as a unit, called one PASS. In our experimental setting, the number of exemplars $K = 8$ for all datasets other than E-SNLI where $K = 9$, the size of development set $M = 256$, the typical number of candidate explanations in $\hat{E}_i$, marked as $|\hat{E}|$, for each question is 8. We will use $K = 8$, $|\hat{E}| = 8$ for estimating the overhead. Scoring one combination with $\mathcal{O}$ requires processing $M(K+1) = 2304$ number of examples.

To compute $\mathcal{S}_{\text{OSLL}}$ for all combinations, we need to score all pairs of $e_i$ and $e_j$ where $e_i \in \hat{E}_i \wedge e_j \in \hat{E}_j \wedge i \neq j$ by $p(a_i, e_i, q_i \mid a_j, e_j, q_j; \theta)$. In total, the overhead involves processing $2|\hat{E}|^2 K(K-1) = 7168$ number of examples. The computation cost is equivalent to scoring 3.1 combinations against silver set.

The overhead for $\mathcal{S}_{\text{OSAcc}}$ requires performing one-shot inference for all explanation candidates, which process $2|\hat{E}|KM = 32768$ examples. The overhead is equivalent to scoring 14.2 combinations.

Note that this computation just needs to be performed once for each task. If we are deploying a system in practice, we ideally want to find one strong prompt that can work well for the task. These expenses are analogous to the training phase for fine-tuned models, and are small compared to the overall cost to do inference on a high number of examples in a real system.

**Details of computation budget**  We now detail how the budget $B$ is allocated to computing the proxy metrics and scoring combinations using $\mathcal{O}$. Consider computation budget of 50 as used in the main experiments (Section 7.1). As discussed before, the overhead for computing $\mathcal{S}_{\text{OSLL}}$ for all combinations is roughly equivalent to 3 PASSES; the overhead for $\mathcal{S}_{\text{OSAcc}}$ is roughly 14 PASSES. Therefore, we allow ENSEMBLE to rank 32 combinations in total (16 from $\mathcal{S}_{\text{OSLL}}$ and 16 from $\mathcal{S}_{\text{OSAcc}}$). For the reduced budget setting that sets $B$ to be 20, we only allow ENSEMBLE to rank 2 combinations, one from $\mathcal{S}_{\text{OSLL}}$ and one from $\mathcal{S}_{\text{OSAcc}}$.

## D  Additional Analysis on Proxy Metrics

|  | GSM | ECQA | E-SNLI | STRQA |
|---|---|---|---|---|
| NAIVE | 64.6 | 79.8 | 82.1 | 70.7 |
| $\mathcal{S}_{\text{OSLL}}$ | 64.7 | **82.7** | 80.6 | **71.8** |
| $\mathcal{S}_{\text{OSAcc}}$ | 65.7 | 81.9 | **83.3** | 70.7 |
| ENSEMBLE | **66.0** | 82.5 | 83.0 | 71.6 |

Table 9: Comparing the performance of different proxy metrics. $\mathcal{S}_{\text{OSLL}}$ and $\mathcal{S}_{\text{OSAcc}}$ are more effective than NAIVE. ENSEMBLE is the best overall.

**Setup**  To give further evidence on the effectiveness of using our proxy metrics, we evaluate the performance of explanations obtained using different proxy metrics, and compare against NAIVE that chooses random combinations. We show the results in Table 9. Note that all approaches use the same amount of computation budget (50) to ensure fair comparison. Specifically, we allow NAIVE to rank 50 combinations, $\mathcal{S}_{\text{OSLL}}$ to rank 48 combinations, $\mathcal{S}_{\text{OSAcc}}$ to rank 32 combinations, and ENSEMBLE to rank 32 combinations (16 of each); this roughly equalizes the overall computation needed for each approach.

**Results**  As shown in Table 9, applying our approach in a NAIVE way can already lead to accuracy improvements compared to the seed set. Under the same computation budget, using proxy metrics to prioritize search strategy can further improve the performance of the searched explanations, compared to NAIVE. $\mathcal{S}_{\text{OSLL}}$ is especially effective on ECQA, whereas $\mathcal{S}_{\text{OSAcc}}$ achieves the best performance

on E-SNLI. Using an ensemble of the two strategies leads to the best overall performance, improving performance compared to NAIVE across all datasets.

## E  Varying Explanations versus Varying Order

|  | Min | Avg | Max |
|---|---|---|---|
| GSM | 58.0 | 61.5 | 64.6 |
| ECQA | 71.8 | 73.9 | 76.2 |
| E-SNLI | 68.7 | 73.7 | 76.2 |
| STRATEGYQA | 70.5 | 74.2 | 76.8 |

Table 10: Statistics of the performance of 16 different random order on four datasets. Varying order has less impact compared to varying explanations (Table 1).

Given a set of exemplars, our approach optimizes the downstream performance by optimizing explanations. Past work has suggested different order of exemplars can also lead to variance in downstream performance (Lu et al., 2022).

We find that varying explanations has a larger impact than varying order. We compare the potential headroom that could be achieved by optimizing explanations against optimizing order. As in Table 10, we show the statistics of the performance of 16 different random orders of the seed explanations, with a similar setup as Table 1 in the main paper. We can conclude that on GSM, ECQA, E-SNLI, the best prompts (MAX) that we can find by varying order are less effective than varying explanations (see Table 1).

## F  Significance Test on the Main Results

|  | GSM | ECQA | E-SNLI | STRQA |
|---|---|---|---|---|
| SEED | 62.6 | 77.0 | 75.2 | 71.3 |
| OPTIMIZED | 66.0 | 83.0 | 82.8 | 71.6 |
| Significance | ⇑⇑⇑↑ | ⇑⇑⇑⇑ | ⇑⇑⇑⇑ | −−↓− |

Table 11: Significance test on the comparison between OPTIMIZED explanations and SEED explanations. The gain is typically significant.

Recall that we experiment with 4 random trials, varying different sets of exemplars $T$ and development sets $V$ in our main experiments (Section 7.1). Therefore, for comparison on each dataset, we perform 4 paired tests, checking whether the improvements of OPTIMIZED over SEED are significant in each setting. We use one ⇑ and one ↑ to denote OPTIMIZED is significantly better than SEED with a p-value < 0.05 and 0.1, respectively. We use ↓ and ⇓ to denote the OPTIMIZED is significantly worse, and use − to denote there are no significant differences. As shown in Table 11, our optimization approach almost never downgrades the performance (except for one trial on STRATEGYQA). And on GSM, ECQA, and E-SNLI, the gain is typically significant with a p-value < 0.05.

## G  Output Examples

We show an example of optimized explanations we get using ENSEMBLE and computation budget $B = 50$ on GSM, ECQA, E-SNLI, and STRATEGYQA in Figure 5, Figure 6, Figure 7, and Figure 8, respectively. In addition, we manually check the 32 (4 trials * 8 examples) optimized explanations for GSM, and 96.9% of them are valid. Our approach typically generates sound explanations for the task of arithmetic reasoning.

| Example outputs on GSM |
|---|

**Q:** Bob has planted corn in his garden, and it has just started to sprout. A week after planting it, it had grown 2 inches. The next week, its height increased by twice as much as it had the first week. In the third week, it grew 4 times as much as it did the week before. How tall are the corn plants now?

**Seed:** The second week it grew twice as much as the first week, so 2 * 2 inches = 4 inches. The third week it grew 4 times as much as in the second week, so 4 * 4 inches = 16 inches. In total, it grew 2 inches + 4 inches + 16 inches = 22 inches. The answer is 22.

**Searched:** After growing 2 inches in the first week, the corn is 2 inches tall. After growing 2 x 2 = 4 inches in the second week, the corn is 2 + 4 = 6 inches tall. After growing 4 x 4 = 16 inches in the third week, the corn is 6 + 16 = 22 inches tall. The answer is 22.

**Q:** Benjamin is tracking how many miles he walks in a week. He walks to work and home five days a week, walks his dog twice a day every day, walks to his best friend's house once a week, and walks to the convenience store twice a week. Work is six miles away, dog walks are two miles, his best friend's house is one mile, and the convenience store is three miles. How many miles does Benjamin walk in a week?

**Seed:** Benjamin walks 6 miles to work and 6 miles home, so he walks 6 * 2 = 12 miles to work and home. He walks to work 5 days a week, so he walks 12 * 5 = 60 miles to work and home a week. He walks his dog 2 times a day for 2 miles, so he walks 2 * 2 = 4 miles a day for his dog. He walks his dog 7 days a week, so he walks 4 * 7 = 28 miles for his dog. He walks 3 miles to the store 2 times a week, so he walks 2 * 3 = 6 miles to the store each week. Adding his weekly 1 mile to his friend's house, Benjamin walks 60 + 28 + 6 + 1 = 95 miles in a week. The answer is 95.

**Searched:** Benjamin walks to work and back two times a day, five days a week. That is 2 x 5 x 6 = 60 miles. He also walks his dog twice a day every day. That is 2 x 2 x 7 = 28 miles. He walks to his best friend's house once a week, which is 1 x 1 = 1 mile. Finally, he walks to the convenience store twice a week, which is 2 x 3 = 6 miles. Therefore, Benjamin walks a total of 60 + 28 + 1 + 6 = 95 miles in a week. The answer is 95.

**Q:** In a basketball game, Tobee scored 4 points. Jay scored 6 more than Tobee and Sean scored 2 less than the points of Tobee and Jay together. If Tobee, Jay, and Sean are on the same team, how many points did they score for their team?

**Seed:** Jay scored 4 + 6 = 10 points. Together, Tobee and Jay scores 4 + 10 = 14 points. So, Sean scored 14 - 2 = 12 points. Thus, Tobee, Jay, and Sean scored a total of 4 + 10 + 12 = 26 points for their team. The answer is 26.

**Searched:** Tobee scored 4 points. Jay scored 4 + 6 = 10 points. Tobee and Jay scored 4 + 10 = 14 points. Sean scored 2 less than Tobee and Jay, so he scored 14 - 2 = 12 points. Tobee, Jay, and Sean scored a total of 4 + 10 + 12 = 26 points. The answer is 26.

**Q:** Bobby made a big batch of soup weighing 80 kg. Each day the batch gets reduced by half. How many kg will it reach on the fourth day after he made the soup?

**Seed:** On the first day, the soup reduces down to 80 / 2 = 40 kg. On the second day, it reduces down to 40 / 2 = 20 kg. On the third day, it is reduced to 20 / 2 = 10 kg. And on the fourth day, it is finally down to 10 / 2 = 5 kg. The answer is 5.

**Searched:** On the first day, there are 80 - 80/2 = 80 - 40 = 40 kg of soup. On the second day, there are 40 - 40/2 = 40 - 20 = 20 kg of soup. On the third day, there are 20 - 20/2 = 20 - 10 = 10 kg of soup. On the fourth day, there are 10 - 10/2 = 10 - 5 = 5 kg of soup. The answer is 5.

**Q:** Rob and Mark plan to meet at the national park. It takes 1 hour for Rob to get to the national park and it takes three times as much time for Mark to get to the national park. If Rob leaves his home at 11 a.m., at what time should Mark leave his home so that they both arrive at the same time?

**Seed:** It takes 1 hour * 3 = 3 hours for Mark to get to the national park. The extra time it takes Mark to get to the national park is 3 hours - 1 hour = 2 hours. Mark should leave his home at 11 a.m. - 2 hours = 9 a.m. The answer is 9.

**Searched:** It takes 1 hour for Rob to get to the park. It takes 3 hours for Mark to get to the park. So, Rob gets to the park at 12 noon (11 a.m. + 1 hour). If Mark wants to get to the park at the same time as Rob, he would have to leave his home at 9 a.m. (12 noon - 3 hours). The answer is 9 a.m.

**Q:** Yanna baked twenty butter cookies and forty biscuits in the morning. In the afternoon, she baked ten butter cookies and twenty biscuits. How many more biscuits did she bake than butter cookies?

**Seed:** There were 20 + 10 = 30 butter cookies. And, there were 40 + 20 = 60 biscuits. Therefore, she baked 60 - 30 = 30 more biscuits than butter cookies. The answer is 30.

**Searched:** In the morning, Yanna baked 20 butter cookies and 40 biscuits. In the afternoon, she baked 10 butter cookies and 20 biscuits. She baked a total of 20 + 10 = 30 butter cookies, and 40 + 20 = 60 biscuits. She baked 60 - 30 = 30 more biscuits than butter cookies. The answer is 30.

**Q:** After Bella's ballet performance, she received 2 dozen roses from her parents, and 2 roses from each of her 10 dancer friends. How many roses did Bella receive?

**Seed:** Bella receives 2 x 12 = 24 roses from her parents. She receives 2 x 10 = 20 roses from her friends. She receives a total of 24 + 20 = 44 roses. The answer is 44.

**Searched:** Bella receives 2 x 12 = 24 roses from her parents. She receives 2 x 10 = 20 roses from her friends. She receives a total of 24 + 20 = 44 roses. The answer is 44.

**Q:** Honey earned $80 a day. Every day, she spent part of her pay and saved the rest. After 20 days of work, she spent $1360. How much did Honey save in 20 days?

**Seed:** Honey earned 20 x $80 = $1600 for 20 days of work. Therefore, Honey saved $1600 - $1360 = $240 in 20 days. The answer is 240.

**Searched:** Honey spent $1360 in 20 days, so she spent $1360 / 20 = $68 a day. She saved $80 - $68 = $12 a day. She saved $12 x 20 = $240. The answer is $240.

Figure 5: Examples of seed explanations and search outputs for GSM.

| Example outputs on ECQA |
|---|
| Q: Sally brought the ball when she visited Scott so that they could do what with it?
Answer Choices: (a) bounces (b) play with (c) toy (d) charming (e) earball
**Seed:** Ball is spherical toy. Toys can be played with. So the answer is (b).
**Searched:** Sally brought the ball when she visited Scott so that they could play with it. So the answer is (b). |
| Q: What are most people trying to do when going on vacation? Answer Choices: (a) panic (b) debate (c) having fun (d) debt (e) peace
**Seed:** Vacation is a holiday during which people relax and enjoy away from home. Relax and enjoy means having fun. So the answer is (c).
**Searched:** Most people go on vacation to have fun. So the answer is (c). |
| Q: The birds were looking around in August, what were they looking to do?
Answer Choices: (a) breathe (b) sky (c) build a nest (d) pick berries (e) reproduce
**Seed:** Birds generally moult in august. Birds want to collect food before they start moulting. Picking berries is the action of collecting food. So the answer is (d).
**Searched:** Birds generally moult in august. Birds want to collect food before they start moulting. Picking berries is the action of collecting food. So the answer is (d). |
| Q: Where could you find a monkey, elephant, or zebra enjoying a drink?
Answer Choices: (a) african continent(b) bar (c) amazon basin (d) research laboratory (e) friend's house
**Seed:** Amazon basin is a big wiuld areas where many animals can be found. Amazon basin has lots of water hence all monkey, elephant, or zebra can be found enjoying it. So the answer is (c).
**Searched:** Amazon basin is a jungle in South America where monkey, elephant and zebra can be found. So the answer is (c). |
| Q: Many in the flock often forget the most basic tenants, a big one being that on god can what?
Answer Choices: (a) anything (b) everywhere (c) judge men (d) give peace (e) help people
**Seed:** Bible says that only god can judge men. Bible is where the most basic tenants are listed. So the answer is (c).
**Searched:** Many in the flock often forget the most basic tenants, a big one being that on god can judge men. So the answer is (c). |
| Q: Food on what kind of transport is normally freeze dried or dehydrated?
Answer Choices: (a) friend's house (b) bicycle (c) space shuttle (d) kitchen (e) disneyland
**Seed:** Space shuttle is type of transportation that astronaut uses. Generally food are freeze dried or dehydrated before taking on space shuttle. So the answer is (c).
**Searched:** Food on space shuttle is normally freeze dried or dehydrated. So the answer is (c). |
| Q: As everybody at the table began to twirl noodles she mentioned that wine was he secret ingredient in the what?
Answer Choices: (a) liquor store (b) church (c) museum (d) spaghetti sauce (e) art show
**Seed:** Spaghetti sauce contains wine as a secret ingredient. As everybody at the table began to twirl noodles she mentioned that wine was her secret ingredient in the spaghetti sauce. So the answer is (d).
**Searched:** Spaghetti sauce contains wine as a secret ingredient. As everybody at the table began to twirl noodles she mentioned that wine was her secret ingredient in the spaghetti sauce. So the answer is (d). |
| Q: The person gave a good hands on what to show how to do the work?
Answer Choices: (a) demonstration (b) garage (c) building (d) barn (e) synagogue
**Seed:** To show how to do something means to give a demostration. So the answer is (a).
**Searched:** Demonstration is a way of showing how to do the work. So the answer is (a). |

Figure 6: Examples of seed explanations and search outputs for ECQA.

| Example outputs on E-SNLI |
|---|
| Premise:
"A young man hitting a ball with a bat."
Based on this premise, can we conclude the hypothesis "Three girls are playing basketball." is true?
**Seed:** There are either three girls or one young man. The answer is no.
**Searched:** The answer is no. |
| Premise:
"A woman is tacking up a sign on the sigh post."
Based on this premise, can we conclude the hypothesis "The woman is eating a cake." is true?
**Seed:** Someone cannot tack a sign and eat cake at the same time. The answer is no.
**Searched:** One cannot be eating cake while tacking up a sign. The answer is no. |
| Premise:
"A mother and her child are out for a walk."
Based on this premise, can we conclude the hypothesis "A mother and her child are bonding together." is true?
**Seed:** going on a walk doesn't imply bonding together. The answer is not possible to tell.
**Searched:** We don't know if they are bonding together. The answer is not possible to tell. |
| Premise:
"Two children in soccer uniforms."
Based on this premise, can we conclude the hypothesis "The two children wear soccer uniforms pafter the game." is true?
**Seed:** You can not infer there is a game. The answer is not possible to tell.
**Searched:** We don't know if the children wear soccer uniforms after the game. The answer is not possible to tell. |
| Premise:
"A man in a red, black and white uniform is pursuing a soccer ball on a grassy field."
Based on this premise, can we conclude the hypothesis "A man is playing soccer." is true?
**Seed:** if your pursuing a soccer ball your also playing soccer. The answer is yes.
**Searched:** If a man is chasing a soccer ball in a field, he is playing soccer. The answer is yes. |
| Premise:
"Two children lying in the snow making snow angels."
Based on this premise, can we conclude the hypothesis "The two children are having fun." is true?
**Seed:** we don't know they are having fun. The answer is not possible to tell.
**Searched:** You can not infer what the children are feeling. The answer is not possible to tell. |
| Premise:
"A man is walking down a pavement path."
Based on this premise, can we conclude the hypothesis "the man is in the shower." is true?
**Seed:** One cannot be walking and in the shower at the same time. The answer is no.
**Searched:** The man can not be in the shower and walk down a path at the same time. The answer is no. |
| Premise:
"Two bicyclists in a race with helmets."
Based on this premise, can we conclude the hypothesis "two guys on bikes." is true?
**Seed:** Bicyclists can be considered as guys on bikes. The answer is yes.
**Searched:** The answer is yes. |
| Premise:
"A group of children is entering an inflatable bounce house."
Based on this premise, can we conclude the hypothesis "bounce house is filled with children." is true?
**Seed:** Children entering a bounce house implies that the house is filled with children. The answer is yes.
**Searched:** We know there are children in the house. The answer is yes. |

Figure 7: Examples of seed explanations and search outputs for E-SNLI.

| Example outputs on STRATEGYQA |
| --- |
| Q: Did Archduke Franz Ferdinand of Austria participate in the Pacific War?
**Seed:** Archduke Franz Ferdinand of Austria was assassinated in 1914. The Pacific War took place between 1941 and 1945. So the answer is no.
**Searched:** Archduke Franz Ferdinand of Austria was assassinated in 1914. The Pacific War took place between 1941 and 1945. So the answer is no. |
| Q: Does Mercury make for good Slip N Slide material?
**Seed:** The Slip N Slide was an outdoor water slide toy. Mercury is a thick liquid at room temperature. Mercury is poisonous and used to kill hatters that lined their hats with the substance. So the answer is no.
**Searched:** Mercury has a melting point of -38.83 °C and a boiling point of 356.73 °C. The temperature range in which a Slip N Slide may be used is between 10 and 30 °C. So the answer is no. |
| Q: Can the Toyota Hilux tip the scales against Mr. Ed?
**Seed:** The current generation of Toyota Hilux weighs at least 4,310 lbs Mr. Ed was portrayed by an adult horse The average adult horse weighs up to 2,000 lbs So the answer is yes.
**Searched:** The Toyota Hilux weighs about 2,500 kg. Mr. Ed weighs about 500 kg. So the answer is yes. |
| Q: Could Lil Wayne's children ride in a Chevrolet Corvette ZR1 together?
**Seed:** Lil Wayne has four children. A Chevrolet Corvette ZR1 has 2 seats. So the answer is no.
**Searched:** Lil Wayne has four children. The Chevrolet Corvette ZR1 has two seats. So the answer is no. |
| Q: Did Woodrow Wilson consider Blacks to be equal members of society?
**Seed:** Woodrow Wilson supported the Ku Klux Klan. The Ku Klux Klan consider Blacks to be inferior. So the answer is no.
**Searched:** Wilson was a proponent of segregation and a supporter of the Ku Klux Klan. He wrote, "The white men were roused by a mere instinct of self-preservation... until at last there had sprung into existence a great Ku Klux Klan, a veritable empire of the South, to protect the Southern country." So the answer is no. |
| Q: Paleography hypothetically helps to understand Cthulhu?
**Seed:** Palaeography involves the study of ancient writings. Paleography has helped decode Anatolian hieroglyphics from all the way back as the first millenia BC. Author H.P. Lovecraft's Cthulhu is an ancient mystical being from eons ago. So the answer is yes.
**Searched:** Paleography is the study of ancient writing. The book, The Call of Cthulhu, was written by H. P. Lovecraft in 1928. So the answer is yes. |
| Q: Could someone listen to the entire U2 debut studio album during an episode of Peppa Pig?
**Seed:** U2's debut studio album was titled Boy. The album, Boy, is 42 minutes and 52 seconds long. An episode of Peppa Pig has a running time of approximately 5 minutes. So the answer is no.
**Searched:** The U2 debut studio album Boy is 38 minutes long. Each episode of Peppa Pig is 5 minutes long. So the answer is no. |
| Q: Are more watermelons grown in Brazil than Antarctica?
**Seed:** Watermelons are plants grown in climates from tropical to temperate, needing temperatures higher than about 25 C (77 F) to thrive. The climate of Antarctica is the coldest on Earth. The climate of Brazil comprises a wide range of weather conditions across a large area and varied topography, but most of the country is tropical. So the answer is yes.
**Searched:** Watermelons are grown in Brazil. There are no watermelons grown in Antarctica. So the answer is yes. |

Figure 8: Examples of seed explanations and search outputs for STRATEGYQA.