# OpenReview forum: "Explanation Selection Using Unlabeled Data for Chain-of-Thought Prompting"
_EMNLP/2023/Conference — EMNLP 2023 Main_

### Official Review · Reviewer_T61D · 2023-07-27

**Typos Grammar Style And Presentation Improvements:** 1. Figure3 show be reworked, the capt…
**Soundness:** 4

**Excitement:**

3: Ambivalent: It has merits (e.g., it reports state-of-the-art results, the idea is nice), but there are key weaknesses (e.g., it describes incremental work), and it can significantly benefit from another round of revision. However, I won't object to accepting it if my co-reviewers champion it.

**Missing References:**

I found that the references about black-box optimization (which heavily rely on prompt optimization/demonstrations) in this paper are to some extent outdated, authors can consider adding the following papers from 2022/23, and discuss how these works construct demos, and their weaknesses.

[1] BBTv2: Towards a Gradient-Free Future with Large Language Models https://aclanthology.org/2022.emnlp-main.259/

[2] When Gradient Descent Meets Derivative-Free Optimization: A Match Made in Black-Box Scenario https://aclanthology.org/2023.findings-acl.55/

[3] Black-box Prompt Learning for Pre-trained Language Models https://arxiv.org/abs/2201.08531

[4] Make Prompt-based Black-Box Tuning Colorful: Boosting Model Generalization from Three Orthogonal Perspectives https://arxiv.org/abs/2305.08088

And the ICL Related papers:

[1] A Survey on In-context Learning https://arxiv.org/abs/2301.00234

[2] Compositional Exemplars for In-context Learning https://arxiv.org/abs/2302.05698

**Paper Topic And Main Contributions:**

Subtly different explanations can largely affect the downstream tasks’ performance when prompting LLMs for reasoning under the CoT paradigm. The paper studies how to optimize explanations by proposing a framework to search for better combinations of explanations when given a set of human-annotated seeds. The authors demonstrate the effectiveness of the method by evaluating it on four popular benchmark reasoning datasets.

**Reasons To Accept:**

1. Authors propose a new framework to find the best in-context explanations for CoT, and use two proxy metrics to approximate downstream performance and enable efficient search. The metrics generally correlate positively with final accuracy.
2. The experiments covered different kinds of reasoning tasks, and the authors provided sufficient details of the statistics. Moreover, the overall presentation of this paper is good.
3. The idea of selecting demonstrations from unlabeled data can be potentially applied to many scenarios.

**Reasons To Reject:**

1. Although the authors use an additional constraint, searching over possible combinations of candidate explanations to find the one with the best performance on dev set might still be very expensive, e.g., paper [1] uses a very similar strategy to (use brute force to) find best in-context demonstrations.
2. The effectiveness of selected explanations is largely dependent on the specific dataset, for instance, the ones for GSM8K dataset can hardly be applied to other benchmarks like Multiarith, SingleEQ/OP, AddSub, GSM-Hard, although they are very similar in topics and have identical input/output formats.

[1] Make Prompt-based Black-Box Tuning Colorful: Boosting Model Generalization from Three Orthogonal Perspectives https://arxiv.org/abs/2305.08088

**Reproducibility:**

4: Could mostly reproduce the results, but there may be some variation because of sample variance or minor variations in their interpretation of the protocol or method.

**Reviewer Confidence:**

4: Quite sure. I tried to check the important points carefully. It's unlikely, though conceivable, that I missed something that should affect my ratings.

---

> ### Author Rebuttal · Authors · 2023-08-28
>
> Thanks for your thoughtful comments and feedback. We will improve the paper as suggested and include more updated references on black-box optimization.
>
>
> **Q1: Although the authors use an additional constraint, searching over possible combinations of candidate explanations to find the one with the best performance on dev set might still be very expensive, e.g., paper [1] uses a very similar strategy to (use brute force to) find best in-context demonstrations.**
>
> A: We acknowledge that our method does not guarantee finding the combination with optimal silver accuracy, especially as we are limiting our computation budget and operating in the black-box setting.
>
> However, we are not aware of any method that can search over the space of prompts for blackbox LMs and find a provably optimal prompt. Other approaches that make different approximations in past work (e.g., RLPrompt) are suitable for more exhaustively searching over a smaller set of options (RLPrompt searches over prompts that are just a few tokens long). Our tradeoff reflects the practical constraints of this complex setting.
>
> **Q2: The effectiveness of selected explanations is largely dependent on the specific dataset, for instance, the ones for GSM8K dataset can hardly be applied to other benchmarks like Multiarith, SingleEQ/OP, AddSub, GSM-Hard, although they are very similar in topics and have identical input/output formats.**
>
> A: This is a great point. We do optimize explanations based on silver accuracy in a particular domain, so our approach is most suitable for tuning a method in that domain. However, we still expect the performance improvements to be able generalize to datasets of somewhat similar distribution.
>
> We test the performance of seed explanations and the optimized explanations on the other arithmetic reasoning datasets as suggested by the reviewer. We note that we only use the validation set from the GSM dataset to optimize the explanations. As shown in the table below, the optimized explanations achieve better performance compared to seed explanations on the out-of-domain datasets, which indicates that the performance improvements can generalize.
>
>
> |           | SVAMP | SingleEq | SingleOp | AddSub | MultiArith |
> |-----------|-------|----------|----------|--------|------------|
> | Seed      | 73.0  | 92.8     | 91.5     | 86.7   | 95.0       |
> | Optimized | 76.9  | 93.4     | 92.2     | 89.6   | 95.6       |
>
>
> We can include these results and discussion in any future version.

---

### Official Review · Reviewer_j9eL · 2023-07-28

**Typos Grammar Style And Presentation Improvements:** N/A
**Soundness:** 4

**Excitement:**

4: Strong: This paper deepens the understanding of some phenomenon or lowers the barriers to an existing research direction.

**Missing References:**

N/A

**Paper Topic And Main Contributions:**

This paper proposes a framework for selecting explanations from a candidate set in a chain-of-thought-style prompting scheme. On a high level, this framework uses proxy metrics to rank generated explanations and these selected explanations lead to consistent performance improvements on 4 datasets on GPT-3 models.

The main contributions are:
- A formulation of the explanation selection problem, and empirically demonstrating that the choice of explanations lead to substantial variation in the performance of a prompted language model
- Efficient metrics that uses one-shot accuracy / log likelihood as proxies for the actual performance of a candidate set of explanations, and well-designed experiments to show that these metrics identify better sets of explanations than random chance
- A framework that selects a set of explanations that lead to substantially better performance than the original set

**Questions For The Authors:**

Why aren't the seed explanations included in the candidate explanation sets?

**Reasons To Accept:**

- The experiment design is solid and there are strong results that demonstrate the effectiveness of proposed metrics and framework.
- It is interesting to see that both proposed metrics $\mathcal{S}_\text{OSAcc}$ and  $\mathcal{S}_\text{OSLL}$ select better explanations than random chance. Figure 3 is informative for understanding why these metrics help.
- The authors provide comprehensive analyses of the framework. The analyses demonstrate that the proposed framework is compatible with a common prompting technique (self-consistency), and works well under a reduced computation budget.

**Reasons To Reject:**

- The proposed method makes use of a silver set of pseudo-labeled examples, and it seems fair to further consider a best-of-$k$ baseline, which samples $k$ candidate explanation sets and then chooses the best set based on its accuracy on the pseudo-labeled examples. This baseline is known to be effective in https://arxiv.org/abs/2112.09332.
- To make searching over candidate explanations tractable, the framework uses a greedy proxy metrics that assume combining the best individual explanations lead to the best set of explanations. This assumption limits the performance of the method (i.e., the optimal set of explanations could be unattainable).
- There are a few moving parts in the proposed framework (e.g., a pseudo-labeled silver set, ensemble of proxy metrics and candidate explanation sets). This complexity could make it difficult to apply this framework in a real-world setting.

**Reproducibility:**

4: Could mostly reproduce the results, but there may be some variation because of sample variance or minor variations in their interpretation of the protocol or method.

**Reviewer Confidence:**

4: Quite sure. I tried to check the important points carefully. It's unlikely, though conceivable, that I missed something that should affect my ratings.

---

> ### Author Rebuttal · Authors · 2023-08-28
>
> Thanks for your thoughtful comments and feedback. We will improve the paper as suggested. Please find the answers below.
>
> **Q1: The proposed method makes use of a silver set of pseudo-labeled examples, and it seems fair to further consider a best-of-k baseline, which samples candidate explanation sets and then chooses the best set based on its accuracy on the pseudo-labeled examples.**
>
> A: Thanks for your question! Randomly sampling candidate explanations is what we call the Naive strategy (in contrast to our more advanced strategies that are based on proxy metrics). We have experimented with this strategy as a baseline in Table 2 and Appendix D.
>
> Please refer to Appendix D for the analysis on the advantages of using proxy metrics over using the naive strategy. The results show using proxy metrics to prioritize search strategy can lead to better optimized explanations compared to searching over random combinations.
>
> **Q2: To make searching over candidate explanations tractable, the framework uses greedy proxy metrics that assume combining the best individual explanations lead to the best set of explanations. This assumption limits the performance of the method (i.e., the optimal set of explanations could be unattainable).**
>
> A: We acknowledge that our method does not guarantee finding the combination with optimal silver accuracy, especially as we are limiting our computation budget and operating in the black-box setting.
>
> However, we are not aware of any method that can search over the space of prompts for blackbox LMs and find a provably optimal prompt. Other approaches that make different approximations in past work (e.g., RLPrompt) are suitable for more exhaustively searching over a smaller set of options (RLPrompt searches over prompts that are just a few tokens long). Our tradeoff reflects the practical constraints of this complex setting.
>
>
> **Q3: There are a few moving parts in the proposed framework (e.g., a pseudo-labeled silver set, ensemble of proxy metrics and candidate explanation sets). This complexity could make it difficult to apply this framework in a real-world setting.**
>
> A: We disagree a bit with this interpretation. The core of our approach only involves scoring combinations of explanations as opposed to a sophisticated tuning process. We designed our workflow to only require the ability to do inference under models, and to only use a limited number of inference calls to keep the cost low. So we believe this is a realistic framework for optimizing explanations that *could* be used in practice, more so than other methods in this space.
>
> **Q4: Are the seed explanations included in the candidate explanation sets?**
>
> A: Yes. They are included in the candidate explanation sets. We will make this clearer in any future version.

---

### Official Review · Reviewer_cVLb · 2023-08-06

**Soundness:** 4

**Excitement:**

4: Strong: This paper deepens the understanding of some phenomenon or lowers the barriers to an existing research direction.

**Missing References:**

https://arxiv.org/abs/2102.07350
https://aclanthology.org/2022.findings-acl.50/

**Paper Topic And Main Contributions:**

This paper analyzes an interesting observation that different explanations in chain of thought can produce different accuracy. For example,
explanations written by non-experts may lead to lower performance. The authors tackle the problem of how to optimize explanation-infused prompts in a black-box fashion. They show that pseudo labeling an unlabeled dataset can be used to evaluate such combinations. Across four textual reasoning tasks spanning question answering, mathematical reasoning, and natural language inference, they show the efficacy of their method.

**Questions For The Authors:**

See above

**Reasons To Accept:**

- interesting and timely problem statement.
- reasonable solution (I like the emphasis and discussion on computational budget)
- well written and interesting analysis.
- by products like proxy metrics, pseudo labelling of unlabeled data can be useful for other applications.

**Reasons To Reject:**

The following are the questions I have and these are not necessarily 'reasons to reject'.
- I was looking for a comparison with the zero-shot chain of thought baseline which authors refer as ZOT (Kojima et al., 2022). The example selection method has a cost. Also, few shot experiments involve extra token usage cost than zero shot.

- Some of the numbers while comparing proposed method vs baselines seem to be pretty close. Wondering, if authors did any statistical significance test?

- A parallel field to explanation selection is prompt/instruction engineering, where we often change the zeroshot instruction. Another alternative is prompt-tuning via gradient descent. Wondering if authors have any thoughts regarding the tradeoff.

- Few shot examples has various types of example biases such as majority bias, recency bias etc. (http://proceedings.mlr.press/v139/zhao21c/zhao21c.pdf, https://aclanthology.org/2023.eacl-main.130/, https://aclanthology.org/2022.acl-long.556.pdf). Wondering if authors have any thought on how the robustness look like with the application of their method?

I am looking forward to hear answers to these questions from the authors.

**Reproducibility:**

4: Could mostly reproduce the results, but there may be some variation because of sample variance or minor variations in their interpretation of the protocol or method.

**Reviewer Confidence:**

4: Quite sure. I tried to check the important points carefully. It's unlikely, though conceivable, that I missed something that should affect my ratings.

---

> ### Author Rebuttal · Authors · 2023-08-28
>
> Thanks for your thoughtful comments and feedback. We will improve the paper as suggested and include any missing references.
>
>
> **Q1: I was looking for a comparison with the zero-shot chain of thought baseline which authors refer as ZOT (Kojima et al., 2022). The example selection method has a cost. Also, few shot experiments involve extra token usage cost than zero shot.**
>
> A: Our comparisons focus on the few-shot setting, since zero-shot CoT generally under-performs few-shot prompting with manually written explanations (Kojima et al., 2022).
>
> In the table, we show the Zero-Shot CoT performance of code-davinci-002 on the four datasets we use. We note that the results are reproduced using the official code provided by Kojima et al. (2022). As we can see, the performance of zero-shot CoT is far behind few-shot CoT using crowdworker explanations (and few-shot CoT using our optimized explanations;, see Table 3 of our paper). This is also congruent with the findings of prior work (Kojima et al., 2022; Wei et al., 2022b; Zhang et al., 2023). We can include these results and discussion in any future version.
>
> |          | GSM  | ECQA | ESNLI | StrategyQA |
> |----------|------|------|-------|------------|
> | zero CoT | 30.9 | 61.2 | 49.7  | 55.1       |
> | Seed     | 62.8 | 77.0 | 75.2  | 71.3       |
>
>
> Our method is indeed more expensive than zero-shot CoT, but we feel the performance differences put it in a different regime entirely in terms of performance, hence the lack of a more detailed comparison.
>
> **Q2: Wondering if authors did any statistical significance test?**
>
> A: Yes, we have performed significance tests to verify whether the optimized explanations are better than seed explanations. The gain is typically significant. Please refer to Appendix F and Table 5 for details.
>
> **Q3: A parallel field to explanation selection is prompt/instruction engineering, where we often change the zeroshot instruction. Another alternative is prompt-tuning via gradient descent. Wondering if authors have any thoughts regarding the tradeoff.**
>
> A: We believe that using better templates (instructions) optimized by the prompt tuning techniques would bring orthogonal benefits to our approach. An optimized template would produce more accurate silver labels, which allows selecting better combinations.
>
> Our experiments are based on the commonly used prompt templates from past work, which have already been shown to achieve strong performance (Wei et al., 2022; Wang et al., 2022b).
>
> **Q4: Few-shot examples have various types of example biases such as majority bias, recency bias etc. Wondering if authors have any thoughts on how robustness looks like with the application of their method?**
>
> A: Great question! While we didn’t test it explicitly, our method is able to combat these biases. Because we use the accuracy on a silver-labeled validation set to optimize explanations, our methods can filter out combinations that would lead to highly biased predictions marked by poor silver accuracy.
>
> In particular, our approach is robust to annotation bias, as suggested by the fact that we can improve explanations that are actually annotated by crowdworkers (we use crowdsourced seed explanations throughout our experiment). In addition, we find optimizing explanations is more effective than optimizing the order of examples as discussed in Appendix E.

---

### Meta-Review · Area_Chair_nRqo · 2023-09-17

**Recommendation:** 5

**Metareview:**

The paper discusses an approach to get candidate explanations of data for use in the chain of thought prompting, starting with crowd-sourced seed explanations set. The proposed approach efficiently searches over the set of explanations using proxy/surrogate metrics to find a suitable combination to be used as few-shot examples.

Reviewers appreciated the clarity of the paper and experiments on multiple datasets.
All the reviewers found the paper to be sound and interesting.
All appreciated that the proposed approach to get explanations from unlabelled data as well as efficiently selecting a useful combination of explanations can be useful in multiple tasks and scenarios.

Reviewers also highlighted that the proposed techniques do not guarantee that the optimal combination would be selected. Authors should consider clearly specifying this limitation in any future version of their paper.

---

### Decision · Program_Chairs · 2023-10-07

**Decision:**

Accept-Main

**Comment:**

The paper discusses an approach to get candidate explanations of data for use in the chain of thought prompting, starting with crowd-sourced seed explanations set. The proposed approach efficiently searches over the set of explanations using proxy/surrogate metrics to find a suitable combination to be used as few-shot examples.

Reviewers appreciated the clarity of the paper and experiments on multiple datasets.
All the reviewers found the paper to be sound and interesting.
All appreciated that the proposed approach to get explanations from unlabelled data as well as efficiently selecting a useful combination of explanations can be useful in multiple tasks and scenarios.

Reviewers also highlighted that the proposed techniques do not guarantee that the optimal combination would be selected. Authors should consider clearly specifying this limitation in any future version of their paper.